# Parental, pregnancy and neonatal characteristics during the perinatal period as potential risk factors for childhood cancer: FeToxCancer case-control study

Anja Stajnko[1], Jesse Daniel Thacher[1], Anna Oudin[2], Christian Lindh[1], Thomas Lundh[1], Ingrid Øra[3,4], Jenny Selander[5], Lars Rylander[1], Maria Albin[5], Karin Källén[1], Karin Broberg [1,5]*

1 Division of Occupational and Environmental Medicine, Department of Laboratory Medicine, Lund University, Lund, Sweden, 2 Division of Occupational and Environmental Medicine, Department of Clinical Sciences Lund, Orthopaedics, Lund University, Lund, Sweden, 3 Clinical Sciences, Paediatric Oncology, Lund University, Lund, Sweden, 4 Karolinska University Hospital, Stockholm, Sweden, 5 Institute of Environmental Medicine, Karolinska Institutet, Stockholm, Sweden

* karin.broberg@med.lu.se

## Abstract

Childhood cancer aetiology is poorly understood and is considered to originate in utero and early postnatal life. In this study, we investigated perinatal characteristics as potential risk factors by performing a population-based case-control study, including 1340 cancer cases diagnosed < 19y and born between 1989–2021 in southern Sweden, and 13400 controls matched by sex, year, and municipality of birth. Perinatal characteristics were obtained from seven national registries. Cox regression was used to examine the associations between perinatal characteristics and the risk of overall childhood cancer, leukaemia, CNS tumours, lymphoma, and other cancer types combined (OCT). Large for gestational age was associated with a higher risk of overall cancer (HR, 95%CI: 1.32, 1.02–1.69) and leukaemia (HR, 95%CI: 1.58, 1.01–2.5), while a 5-min Apgar score <7 indicated a higher risk of OCT (HR, 95%CI: 2.16, 1.12–4.15). Mechanical ventilation during neonatal care was associated with a higher risk of overall cancer (HR, 95%CI: 1.88, 1.39–2.53) and OCT (HR, 95%CI: 2.09, 1.19–3.39). The aforementioned characteristics were associated with up to a threefold increased risk among children diagnosed before six months of age compared to those diagnosed later. Additionally, maternal obesity was associated with a higher risk of CNS tumours (HR, 95%CI: 1.51, 1.04–2.21) and lymphoma (HR, 95%CI: 2.26, 1.31–3.88), and maternal underweight with a higher risk of leukaemia (HR, 95%CI: 2.43, 1.40–4.22). Planned caesarean delivery indicated an increased risk of OCT (HR, 95%CI: 1.52, 1.04–2.22). Our findings identify several perinatal characteristics associated with childhood cancer risk, highlighting the perinatal period as an important window for future etiological research.

**Data availability statement:** Due to the personal data collected and potentially identifying information I contained within the data, the data are available upon request. Ethical approval by the Swedish Ethical Review Authority may be necessary, and requests for data may be sent to: SOCIALSTYRELSEN Statistikavdelningen Enheten för Mikrodata 106 30 Stockholm email: socialstyrelsen@socialstyrelsen.se.

**Funding:** 1) -KB, -Grant No: Dnr 20 0825 Pj; -full name: Swedish Cancer Society, -URL: https://www.cancerfonden.se/om-oss/about -no 2) -KB; -Grant No: PR2020-0012; -full name: Swedish Childhood Cancer Foundation; -URL: https://www.barncancerfonden.se; -no 3) KB; -Grant No: 2022-Projekt0162; -Full name: ALF; -url: https://www.intramed.lu.se/en/research/alf-funding; -no 4) -KB, -Grant No: 2022-01-11:10, -full name: Sjöberg Foundation, -URL: https://sjobergstiftelsen.se/ -no.

**Competing interests:** The authors have declared that no competing interests exist.

## Introduction

Cancer in early and late childhood is a rare condition and one of the leading causes of mortality among children in high-income countries. The Swedish Children's Cancer Registry reported an average of 330 incident childhood cancer cases annually over the past decade [1], and around 400,000 annual incident cases are estimated worldwide [2]. While global childhood cancer incidence is increasing [3], Sweden`s rate has remained constant over the past decade [1]. The most frequently occurring childhood cancer types are leukaemia, brain and other central nervous system (CNS) tumours, lymphoma, and Wilms tumour [1,2]. Contrary to cancer in adults, the causes of childhood cancers are largely unknown. Investigations of the underlying genetic background have shown that, on average, only 7–10% of childhood cancer cases are carriers of germline mutations in cancer-predisposing genes [4–6]. Most childhood cancer cases likely result from de novo mutations in genes leading to uncontrolled cell growth and cancer. Given that approximately half of the childhood cancer diagnoses occur before the age of five, [1,7] the initial genetic mutations likely originate *in utero* and in early postnatal life [8,9]. Accordingly, multiple epidemiological studies have investigated several perinatal factors, including demographic, environmental, and intrinsic factors, as risk factors for childhood cancer [10–13]. Among these, more established risk factors are exposure to high-dose ionising radiation, prior chemotherapy, age, sex, and ethnicity (i.e., attributable to both genetic and environmental factors) [4,13]. Nevertheless, the list of perinatal characteristics as potential risk factors is growing. To name a few: maternal smoking, maternal BMI and age, birth weight and size, Apgar score, neonatal treatment, preterm birth, caesarean birth, and parental occupational exposure to chemicals [11,14–19]. However, evidence for most of these remains inconsistent and inconclusive [4,10]. Therefore, additional studies on perinatal factors are crucial to identify those with the highest risk for childhood cancer development.

We aimed to investigate associations between multiple parental, pregnancy, and neonatal characteristics during the perinatal period and the risk of overall childhood cancer, as well as specific cancer groups (i.e., leukaemia, CNS tumour, lymphoma, and other cancer types combined).

## Materials and methods

### Study design and population

This study is a part of the ongoing FeToxCancer project focusing on children born in Southern Sweden. Its main objective is to understand the effects of chemical exposure in utero and possible underlying mechanisms in the aetiology of childhood cancer. The study uses a population-based case-control design, comprising in Phase I extensive registry data (described here) and in Phase II measurements of inorganic and organic toxicants in biobanked maternal serum samples collected during infection screening in early pregnancy (2–14 weeks; presented in a separate paper) (Fig 1).

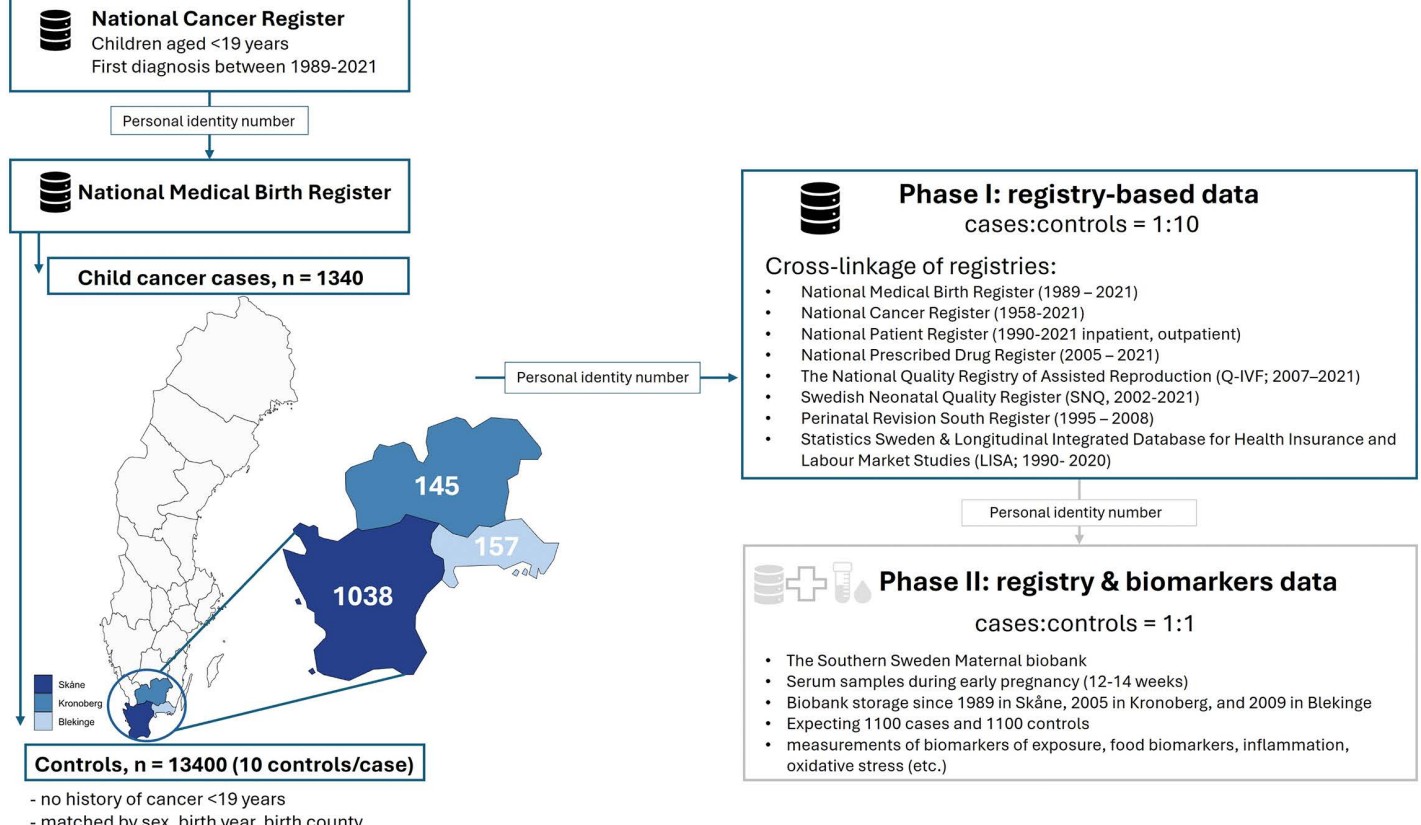

**Fig 1. Flow chart of the FeToxCancer study design.** (in grey: phase II is ongoing and therefore not a part of this study; the map was produced using R programme (version 4.2.3) with Swedish county boundaries obtained from Eurostat's GISCO database [23]).

Serum samples have been stored only within Southern Sweden (i.e., municipalities Skåne, Blekinge and Kronoberg) and are biobanked within the Southern Sweden Maternal Biobank since 1989; thus, this determined the time period and geographic location for the study population.

The FeToxCancer project was approved by the Swedish Ethical Review Authority (No. 2021–02001 and 2021–04399). The need for written consent from the participants was waived by the ethics committee.

Every Swedish resident is assigned a unique personal identification number (i.e., PIN). A valid PIN, expected for nearly 100% of mothers [20], was the primary inclusion criterion and enabled the cross-linkage of various population and health-related national registries (Fig 1). Firstly, all childhood cancer cases with the first diagnosis at age < 19 years and born between 1.1.1989–31.12.2021 (i.e., recruitment period) were selected from the National Cancer Register (NCR). The NCR was established in 1958, and documents cancer cases identified either clinically, by imaging, or by pathology examination, and has 96% coverage [21]. Secondly, cases were linked with the National Medical Birth Register (MBR), which was established in 1973 and contains high-quality pregnancy-related information for around 98% of deliveries in Sweden [22]. All childhood cancer cases born in the municipalities of Skåne (n = 1038), Blekinge (n = 157), and Kronoberg (n = 145) constituted our final case population (n = 1340; Fig 1). Cancer cases were classified into specific groups according to the International Classification of Childhood Cancer (ICCC, 3rd edition; https://seer.cancer.gov/iccc/iccc-iarc-2017.html).

For each case, ten controls matched by sex, birth year, and municipality of birth, were randomly selected from the MBR (n = 13400; Fig 1) from the same population. The inclusion criteria were being born between 1989 and 2021 with no known cancer diagnosis or recorded death before the age of 19, emigration or end of the study, while allowing for cancer diagnosis at the age of 19 or older.

Finally, various national registries were cross-linked (Fig 1) via the maternal PIN to obtain extensive data covering parental, child, and pregnancy characteristics.

The data (including information with the possibility to identify individual participants) was accessed for research purposes on 21st of March 2023.

## Perinatal characteristics

The perinatal characteristics were selected based on current evidence in the literature [11,14–19] and the availability of data within the FeToxCancer study (Table 1). Investigated perinatal characteristics included: I) parental characteristics

Table 1. National registries used to obtain data for the investigated perinatal characteristics.

| Registry | Investigated perinatal characteristic: | Data availability period |
|---|---|---|
| National Medical Birth Register | Maternal Age | 1989-2021 |
| | Parity (i.e., birth order) | 1989-2021 |
| | Maternal BMI[a] | 1989-2021 |
| | Gestational age (GA) | 1989-2021 |
| | Birthweight (to calculate birthweight for GA) | 1989-2021 |
| | Child`s sex (to calculate birthweight for GA) | 1989-2021 |
| | Maternal smoking in the first trimester | 1989-2021 |
| | [Maternal smoking in gestational week 30-32] | [1990-2021] |
| | Mode of delivery | 1989-2021 |
| | Assisted pregnancy with IVF | 1989-2006 |
| | 5-min Apgar | 1989-2021 |
| | [Gestational diabetes] | [1997-2021] |
| Longitudinal Integrated Database for Health Insurance and Labour Market Studies (LISA) | Paternal birthdate (to estimated paternal age) | 1990-2020 |
| | Maternal education | 1990-2020 |
| | Paternal education | 1990-2020 |
| | [Parents' birth country and nationality] | [1990-2020] |
| The National Quality Register of Assisted Reproduction (Q-IVF) | Assisted pregnancy with IVF | 2007-2021 |
| National Patient Register (in-patient care) | Child infections [Child cancer predisposing syndrome[b]] | 1990-2021 [1990-2021] |
| | [Maternal infection during pregnancy] | [1990-2021] |
| Perinatal Revision South Register | Neonatal care [Mechanical ventilation and steroid treatment] | 1995-2015 [1995-2015] |
| Swedish Neonatal Quality Register (SNQ) | Neonatal care [Mechanical ventilation, steroid treatment, phototherapy, antibiotic therapy, surfactant treatment] | 2002-2021 [2002-2021] |
| National Cancer registry | Maternal cancer | 1989-2021 |

IVF – in vitro fertilisation, BMI – body mass index.

[a]at enrolment into the maternal health care.

[b]for Down syndrome, Neurofibromatosis type 1, Congenital malformation syndromes involving early overgrowth (Beckwith-Wiedemann syndrome), and Von Hippel-Lindau syndrome.

in [] are additionally tested perinatal characteristics, which were not included in the main statistical analyses.

including cancer diagnosis of the mother (yes, no), maternal and paternal age at delivery (<25, 25–34 and ≥35 years), maternal and paternal education (primary, secondary, and postsecondary), birth order (i.e., parity; 1st, 2nd, and ≥3rd), maternal BMI (kg/m$^2$) (<18.5 as underweight, 18.5–24.9 as normal, 25–29.9 as overweight, and ≥30 as obese), maternal smoking during the first trimester (yes, no), II) pregnancy characteristics including assisted pregnancy with in vitro fertilisation (IVF; yes, no) and mode of delivery (vaginal, planned caesarean, emergency caesarean, vaginal with forceps or vacuum); and III) neonatal characteristics including gestational age (<37 weeks as pre-term, 37–41 weeks as at term, and ≥42 weeks as post-term), birthweight according to the gestational age (adequate for gestational age (AGA), small for gestational age (SGA) or large for gestational age (LGA), Apgar score at five minutes of age (i.e., 5-min Apgar; ≥7 as reassuring, <7 as low), any childhood infection requiring specialist in-patient care from birth until the age of one year (yes, no), and admission to the neonatal intensive unit (yes, no).

Parental education represents the highest recorded education based on the available data from 1990 to 2020. Maternal BMI was calculated based on maternal weight and height provided at the first antenatal visit in gestational weeks eight to ten. Gestational age (GA) was determined based on the ultrasound, as a superior measure, and in case of a missing ultrasound, based on the day of the last menstrual period (mostly before 1994). We investigated birthweight for GA rather than birth weight, since it has been previously reported as a superior predictor of childhood cancer risk [24]. Birthweight for GA was estimated based on the combination of gestational age, birth weight, and sex of the child and calculated according to the Swedish reference standard growth curves [25]. It was expressed as a standard deviation (SD) score for GA, with SGA defined as SD <−2, AGA as SD between −2 and 2, and LGA as SD >2. We selected the Apgar score at five minutes as it is reported to be a better predictor of long-term adverse health outcomes than the score at one minute [26]. Childhood infections were defined by diagnosis in the International Classification of Diseases (ICD) using the 9th and 10th revisions (ICD9: 001X -139W and ICD10: A000-B99 for any type of infections), and were, for cancer cases, included only if the infection occurred at least more than one month before diagnosis to exclude infections resulting from cancer.

Additionally, we investigated variables of smoking during gestational weeks 30–32 (yes, no), gestational diabetes (ICD10: O240; yes, no), child being diagnosed with a cancer predisposing syndrome [27,28] (information was available for: Down syndrome, ICD9: 758A and ICD10:Q90; Neurofibromatosis type 1, ICD9: 237H and ICD10: Q850; Congenital malformation syndromes involving early overgrowth, ICD9: 756W, 259W and ICD10: Q873; and Von Hippel-Lindau syndrome, ICD9: 759G and ICD10: Q858) [26,27], parents' birth country and nationality (born to foreign parents, born to Swedish parents, or born to one Swedish and one foreign parent), maternal infection during pregnancy (ICD9: 001X–139W and ICD10: A000–B99; yes, no) and specific neonatal treatments. The above-listed characteristics were not included in the main statistical analyses (described below) due to low statistical power or extensive missing data.

## Statistical analyses

In the present study, follow-up lengths vary considerably, with children born in the later part of the study period having significantly shorter follow-up compared to those born in the early years of the study. Traditional case-control analyses using logistic regression do not allow direct estimation of time-varying associations. Therefore, in the present study, the Cox proportional hazards regression models with child age as the underlying time scale were used to calculate hazard ratios (HR) for the association between perinatal characteristics and risk of childhood cancer. A similar statistical approach was also used by other similar studies [10,14,29]. The associations were examined for overall childhood cancer and separately for leukaemia, CNS tumours, lymphoma, and other cancer types combined (i.e., ICCC groups IV-XII, including unclassified; Table 2). The time scale (in years) was determined from birth date until the first childhood cancer diagnosis or censored at 19 years of age, emigration, or end of the follow-up on 31st of December 2021, whichever occurred first.

**Table 2. Distribution of childhood cancer cases by year of birth, year of diagnosis, age at diagnosis, sex, and cancer type.**

| All childhood cancer cases, N (%) | 1340 (100) | | | | | |
|---|---|---|---|---|---|---|
| **Birth year,** N (%) | | | | | | |
| 1989-1999 | 586 (44) | | | | | |
| 2000-2010 | 537 (40) | | | | | |
| 2011-2021 | 217 (16) | | | | | |
| **Year of first diagnosis,** N (%) | | | | | | |
| 1989-1999 | 191 (14) | | | | | |
| 2000-2010 | 475 (35) | | | | | |
| 2011-2021 | 736 (55) | | | | | |
| **Age at first diagnosis**, N (%) | | | | | | |
| At birth | 19 (1.4) | | | | | |
| ≤6 months | 106 (8) | | | | | |
| ≤ 1 year | 264 (20) | | | | | |
| ≤5 years | 640 (48) | | | | | |
| 6-10 years | 248 (18) | | | | | |
| 11-18 years | 452 (34) | | | | | |
| Average age at diagnosis, $\bar{x}\pm$SD | 7.5±6.0 | | | | | |
| **Sex,** N (%) | | | | | | |
| Male | 7799 (53) | | | | | |
| Female | 6941 (47) | | | | | |
| | **All** | **Sex** (N%)[b] | | **Age at diagnosis** (years; N%)[b] | | |
| | **(N%)[a]** | **Male** | **Female** | **≤5** | **6–10** | **11–18** |
| **ICCC groups:** | | | | | | |
| I Leukaemia | 345 (25) | 192 (56) | 153 (44) | 223 (66) | 62 (18) | 60 (17) |
| II Lymphoma | 152 (11) | 92 (60) | 60 (40) | 47 (31) | 37 (24) | 68 (45) |
| III CNS tumors | 328 (24) | 168 (51) | 160 (49) | 125 (38) | 84 (24) | 119 (36) |
| *IV Neuroblastoma & Other Peripheral Nervous Cell Tumors* | 71 (5) | 47 (66) | 24 (34) | 64 (90) | 5 (7) | 2 (3) |
| *V Retinoblastoma* | 25 (2) | 17 (68) | 8 (32) | 25 (100) | – | – |
| *VI Renal tumors* | 76 (6) | 36 (47) | 40 (53) | 66 (87) | 7 (9) | 3 (4) |
| *VII Hepatic tumors* | 12(1) | 7 (58) | 5 (42) | 12 (100) | – | – |
| *VIII Bone tumors* | 57(4) | 35 (61) | 22 (39) | 6 (11) | 15 (26) | 36 (63) |
| *IX Soft tissue tumors* | 67 (5) | 35(53) | 32 (47) | 31 (46) | 13 (19) | 23 (34) |
| *X Germ-cell, trophoblastic & other gonadal neoplasms* | 72(5) | 40 (56) | 32 (44) | 22 (31) | 8 (11) | 42 (58) |
| *XI Carcinomas & other malignant epithelial neoplasms* | 131(10) | 38 (29) | 93 (71) | 9 (7) | 16 (12) | 106 (81) |
| *XII Other & unspecified malignant neoplasms* | 11(0.8) | 5 (45) | 6 (54) | 7 (64) | 2 (18) | 2 (18) |
| *Not classified by ICCC* | 8(0.6) | 4 (50) | 4 (50) | 6 (75) | 2 (25) | – |

ICCC – International Childhood Cancer classification; CNS – central nervous system; [a]based on all cancer diagnoses (n=1355; 15 children were diagnosed with more than one childhood cancer group); [b]based on all diagnoses within a specific cancer group;

 ICCC groups marked in italic are within the manuscript combined into a group referred to as other cancer types combined.

We completed a crude Cox regression model for all perinatal characteristics separately. Subsequently, models with stepwise adjustments were conducted to identify possible influencing variables. The first model was adjusted for all parental characteristics (i.e., maternal cancer diagnosis, maternal and paternal age and education, parity, maternal BMI and maternal smoking), the second with further adjustment for pregnancy characteristics (i.e., assisted pregnancy with IVF and mode of delivery), and the third, in addition to parental and pregnancy characteristics, further adjustment for GA. This

approach allowed us to investigate the risk of GA as a potential risk factor while also addressing its known potential as a mediating factor that might introduce collider bias [30]. The models were not assessed for characteristics or categories when there were fewer than ten observations among cases or controls [29].

Sensitivity analyses were performed by repeating the crude models for perinatal characteristics in the subset of individuals with complete data for all adjustment variables. Some perinatal characteristics were further analysed by a) excluding cases with a diagnosis at birth, b) excluding children diagnosed with cancer predisposed syndrome (Down syndrome, Neurofibromatosis type1, Congenital malformation syndromes involving early overgrowth, and Von Hippel-Lindau syndrome), c) stratification by age at diagnosis (i.e., ≤6 months, ≤1y or >1y for assessment of LGA and mechanical ventilation risk, while according to the statistical power the stratification of >6 months, 0–5y, 6–10y and 11–18y was used to assesses risk of planned caesarean delivery), d) adjustment for additional covariates to support the interpretation and discussion of the results. The proportional hazards assumption for each variable was tested by statistical tests and graphical diagnostics based on the scaled Schoenfeld residuals. All statistical analyses were performed using R software version 4.2.3 with RStudio version 2023.06.2 using two-tailed tests with a p-value of 0.05 as a criterion for statistical significance.

## Results

In total, 1340 childhood cancer cases born between 1989 and 2021 in Southern Sweden were included. The distribution of cases by year of birth, child`s sex, age at diagnosis, and cancer types is presented in Table 2. The most common cancers were leukaemia (25%), CNS tumours (24%), and lymphoma (11%). There were more male (53%) compared to female (47%) childhood cancer cases. The average age at diagnosis was 7.5 years, with 48% of cases with the first diagnosis at or before the age of 5 years.

The distribution of perinatal characteristics between cases and controls for overall childhood cancer is presented in Table 3 and separately for leukaemia, CNS tumours, lymphoma, and other cancer types combined in S1 Table. Associations between perinatal characteristics and the risk of childhood cancer are shown in Table 4 for overall childhood cancer and summarised in Table 5 for leukaemia, CNS tumours, lymphoma, and other cancer types combined. The complete results are provided in Supporting information files: S2–S5 Tables.

### Associations between neonatal characteristics and risk of childhood cancer

Crude and adjusted models showed a higher risk of overall childhood cancer among children born **LGA** compared to AGA (adjusted HR, 95%CI: 1.32, 1.02–1.69; Table 4). Additionally, LGA was associated with increased risk of leukaemia (adjusted HR, 95%CI: 1.58, 1.01–2.51), and somewhat increased risk for other cancer types combined (adjusted HR, 95%CI: 1.45, 0.98–2.14). Being diagnosed with cancer at birth (S1 Fig), diagnosed with cancer predisposing syndrome (S6 Table), or additional adjustment for gestational diabetes (S7 Table) had no significant effect on observed associations. Stratification by age at diagnosis showed a stronger association for LGA and risk of overall childhood cancer for diagnoses ≤1 year (adjusted HR, 95%CI: 2.06, 1.29–2.31), and particularly ≤6 months (adjusted HR, 95%CI: 3.61, 1.87–6.97) compared to those diagnosed later (adjusted HR, 95%CI: 1.14, 0.84–1.54) (Fig 2).

A **5-min Apgar** score <7 was associated with increased risk of overall childhood cancer and other cancer types combined (crude HR, 95%CI: 1.73, 1.16–2.57; and 2.39,1.38–4.14, respectively). However, after adjustment for pregnancy characteristics and GA, only the association with other cancers combined remained (adjusted HR, 95%CI: 2.16, 1.12–4.15; Tables 4 and 5). Moreover, the latter association remained when excluding cases diagnosed with cancer at birth (S1 Fig), diagnosed with cancer predisposing syndromes (S6 Table), as well as after additional adjustment for admission to neonatal care, and birthweight for GA (Table a in S8 Table). Moreover, the distribution of cancer cases with low 5-min Apgar was higher among those diagnosed ≤6 months or ≤1 year when compared to those diagnosed later (6% 3% and 1.5%, respectively, Table b in S8 Table).

**Table 3.  Distribution of studied perinatal characteristics between cases and controls within FeToxCancer study.**

| Perinatal characteristics | FeToxCancer study | |
|---|---|---|
| | Cases (n = 1340) N(%ᵃ)* | Controls (n = 13400) N(%ᵃ) |
| **Maternal cancer** | | |
| No | 1157 (86) | 11654 (87) |
| Yes | 183 (14) | 1746 (13) |
| **Maternal age (years)** | | |
| <25 | 230 (17) | 2423 (18) |
| 25-34 | 889 (66) | 8849 (66) |
| ≥35 | 221 (17) | 2128 (16) |
| **Paternal age (years)** | | |
| <25 | 96 (7) | 914 (7) |
| 25-34 | 757 (56) | 7762 (58) |
| ≥35 | 475 (35) | 4613 (34) |
| missing | 12 (1) | 111 (1) |
| **Maternal education** | | |
| Primary | 120 (9) | 1319 (10) |
| Secondary | 567 (42) | 5580 (41) |
| Postsecondary | 647 (48) | 6379 (48) |
| missing | 6 (0.4) | 122 (0.9) |
| **Paternal education** | | |
| Primary | 185 (14) | 1832 (14) |
| Secondary | 654 (49) | 6567 (50) |
| Postsecondary | 486 (37) | 4763 (36) |
| missing | 15 (0.2) | 238 (1) |
| **Parity (birth order)** | | |
| 1 | 571 (43) | 5817 (43) |
| 2 | 488 (36) | 4847 (36) |
| ≥3 | 281 (21) | 2736 (21) |
| **Maternal BMI (kg/m²)ᵇ** | | |
| <18.5 | 34 (3) | 294 (2) |
| 18.5–24.9 | 677 (51) | 7016 (52) |
| 25–29.9 | 284 (21) | 2724 (20) |
| ≥30 | 127 (9) | 1186 (9) |
| missing | 218 (16) | 2180 (16) |
| **Maternal smokingᵇ** | | |
| No | 1098 (82) | 11078 (83) |
| Yes | 182 (14) | 1867 (14) |
| missing | 60 (4) | 455 (3) |
| **Assisted pregnancy IVF** | | |
| No | 1314 (98) | 13110 (98) |
| Yes | 26 (2) | 290 (2) |
| **Mode of delivery** | | |
| Vaginal | 1068 (80) | 10896 (81) |
| caesarean elective | 73 (5) | 715 (5) |
| caesarean emergency | 122 (9) | 1043 (8) |
| forceps or vacuum | 77 (6) | 746 (6) |

*(Continued)*

**Table 3.** (Continued)

| Perinatal characteristics | FeToxCancer study | |
|---|---|---|
| | Cases (n = 1340) N(%ᵃ)* | Controls (n = 13400) N(%ᵃ) |
| **GA (weeks)** | | |
| <37 | 100 (7) | 832 (6) |
| 37–41 | 1157 (86) | 11683 (87) |
| ≥42 | 82 (7) | 885 (7) |
| **Birthweight for GAᶜ** | | |
| AGA | 1200 (90) | 12225 (91) |
| SGA | 56 (4) | 541 (4) |
| LGA | 76 (6) | 591 (4) |
| Missing | 8 (0.6) | 43 (0.3) |
| **Child infectionᵈ** | | |
| No | 1247 (93) | 12458 (93) |
| Yes | 42 (3) | 432 (3) |
| missing | 51(4) | 510 (4) |
| **5min-Apgar** | | |
| ≥7 | 1307 (98) | 13195 (99) |
| <7 | 25 (2) | 149 (1) |
| missing | 8 (0.6) | 56 (0.4) |
| **Neonatal careᵉ** | | |
| No | 869 (65) | 9077 (68) |
| Yes | 145 (11) | 1063 (8) |
| missing | 326 (24) | 3260 (24) |

IVF – in vitro fertility.

ᵃbased on non-missing data;

ᵇmaternal body mass index (BMI, kg/m2) and smoking status at enrolment into maternal health care.

ᶜbased on weight, sex, GA and calculated according to Marsál et al., 1996.

ᵈany infection during first year of birth.

ᵉdata available since 2002.

*According to any cancer type, distributions for leukaemia, CNS tumour, lymphoma and other cancer types combined are presented in S1 Table.

Admission into **neonatal care** was associated with a higher risk of overall childhood cancer (adjusted HR, 95%CI: 1.25,1.00–1.56, Table 4) and other cancer types combined (adjusted HR, 95%CI: 1.40, 1.00–1.97, Table 4). Moreover, after the exclusion of cases diagnosed with cancer at birth, only the association with other cancer types combined remained (S1 Fig), while no association remained significant after the exclusion of children diagnosed with cancer predisposing syndromes (S6 Table). Due to low statistical power, we were unable to further explore specific neonatal therapies (Table a in S9 Table), except for mechanical ventilation (Table b in S9 Table). The latter was associated with a higher risk of overall childhood cancer and other cancer types combined (adjusted HR, 95%CI: 1.88, 1.39–2.53 and 2.15, 1.29–3.56, respectively; S9 Table). This association was independent of cancer cases diagnosed at birth (S1 Fig), diagnosis with cancer predisposing syndromes, Apgar score, or birthweight for GA (S9 Table). Moreover, treatment by mechanical ventilation indicated an increased risk of overall cancer, especially among children diagnosed at ≤ 1 year of age (Fig 2).

**Table 4. Associations of perinatal characteristics with the risk of overall childhood cancer.**

| Perinatal characteristics | Crude HR (95%CI) | Crude HR (95%CI) Complete data[a] | 1st Model Adj HR (95%CI) | 2nd Model Adj HR (95%CI) | 3rd Model Adj HR (95%CI) |
|---|---|---|---|---|---|
| **Parental characteristics** | | | | | |
| **Maternal cancer, N** | 14740/1340 | 11816/1078 | 11816/1078 | 11816/1078 | 11816/1078 |
| No | Ref | Ref | Ref | Ref | Ref |
| Yes | 0.99 (0.84, 1.15) | 0.99 (0.83, 1.18) | 1.00 (0.84, 1.19) | 1.00 (0.84, 1.19) | 1.00 (0.84, 1.19) |
| **Maternal age (years), N** | 14740/1340 | 11816/1078 | 11816/1078 | 11816/1078 | 11816/1078 |
| <25 | Ref | Ref | Ref | Ref | Ref |
| 25–34 | 1.08 (0.93, 1.24) | 1.10 (0.94, 1.31) | 1.11 (0.91, 1.35) | 1.10 (0.91, 1.34) | 1.10 (0.91, 1.34) |
| ≥35 | 1.16 (0.97, 1.40) | 1.15 (0.94, 1.42) | 1.11 (0.85, 1.44) | 1.09 (0.83, 1.42) | 1.09 (0.83, 1.42) |
| **Paternal age (years), N** | 14617/1328 | 11816/1078 | 11816/1078 | 11816/1078 | 11816/1078 |
| <25 | Ref | Ref | Ref | Ref | Ref |
| 25–34 | 0.95 (0.76, 1.18) | 0.98 (0.77, 1.26) | 0.91 (0.69, 1.20) | 0.91 (0.69, 1.20) | 0.91 (0.69, 1.20) |
| ≥35 | 1.04 (0.83, 1.29) | 1.10 (0.85, 1.41) | 1.00 (0.74, 1.36) | 1.01 (0.74, 1.36) | 1.01 (0.74, 1.36) |
| **Maternal education, N** | 14612/1334 | 11816/1078 | 11816/1078 | 11816/1078 | 11816/1078 |
| Primary | Ref | Ref | Ref | Ref | Ref |
| Secondary | 1.08 (0.88, 1.32) | 0.98 (0.79, 1.22) | 0.96 (0.77, 1.20) | 0.96 (0.77, 1.21) | 0.96 (0.77, 1.21) |
| Postsecondary | 1.13 (0.93, 1.37) | 1.10 (0.82, 1.26) | 0.98 (0.77, 1.24) | 0.98 (0.77, 1.25) | 0.98 (0.77, 1.25) |
| **Paternal education, N** | 14487/1325 | 11816/1078 | 11816/1078 | 11816/1078 | 11816/1078 |
| Primary | Ref | Ref | Ref | Ref | Ref |
| Secondary | 0.98 (0.83, 1.16) | 1.02 (0.85, 1.22) | 1.01 (0.84, 1.23) | 1.01 (0.84, 1.22) | 1.01 (0.84, 1.22) |
| Postsecondary | 1.05 (0.84, 1.30) | 1.04 (0.86, 1.26) | 1.01 (0.82, 1.25) | 1.02 (0.82, 1.25) | 1.02 (0.82, 1.25) |
| **Parity, N** | 14740/1340 | 11816/1078 | 11816/1078 | 11816/1078 | 11816/1078 |
| 1 | Ref | Ref | Ref | Ref | Ref |
| 2 | 1.03 (0.91, 1.16) | 1.05 (0.91, 1.19) | 1.02 (0.88, 1.13) | 1.03 (0.90, 1.19) | 1.03 (0.90, 1.19) |
| ≥3 | 1.04 (0.89, 1.20) | 1.00 (0.86, 1.18) | 0.95 (0.79, 1.13) | 0.97 (0.81, 1.16) | 0.97 (0.81, 1.16) |
| **Maternal BMI (kg/m²)[b], N** | 12342/1122 | 11816/1078 | 11816/1078 | 11816/1078 | 11816/1078 |
| <18.5 | 1.20 (0.84, 1.71) | 1.20 (0.84, 1.71) | 1.23 (0.86, 1.76) | 1.24 (0.87, 1.77) | 1.23 (0.86, 1.73) |
| 18.5–24.9 | Ref | Ref | Ref | Ref | Ref |
| 25–29.9 | **1.15 (1.03, 1.33)*** | **1.15 (1.00, 1.32)*** | **1.15 (1.00, 1.33)*** | 1.12 (0.96, 1.32) | 1.11 (0.94, 1.37) |
| ≥30 | 1.21 (0.99, 1.47) | 1.21 (1.00, 1.47) | 1.21 (0.99, 1.47) | 1.19 (0.98, 1.44) | 1.19 (0.98, 1.45) |
| **Maternal smoking[b], N** | 14225/1280 | 11816/1078 | 11816/1078 | 11816/1078 | 11816/1078 |
| No | Ref | Ref | Ref | Ref | Ref |
| Yes | 0.90 (0.77, 1.05) | 0.90 (0.75, 1.08) | 0.91 (0.76, 1.10) | 0.91 (0.76, 1.10) | 0.91 (0.76, 1.10) |
| **Pregnancy characteristics** | | | | | |
| **Assisted pregnancy IVF, N*** | 14740/1340 | 11816/1078 | 11816/1078 | 11816/1078 | 11816/1078 |
| No | Ref | Ref | Ref | Ref | Ref |
| Yes | 0.96 (0.65, 1.42) | 0.92 (0.60, 1.41) | 0.87 (0.57, 1.34) | 0.86 (0.56, 1.32) | 0.85 (0.55, 1.31) |
| **Mode of delivery, N*** | 14740/1340 | 11816/1078 | 11816/1078 | 11816/1078 | 11816/1078 |
| Vaginal no instruments | Ref | Ref | Ref | Ref | Ref |
| caesarean elective | 1.11 (0.87, 1.41) | 1.13 (0.87, 1.37) | 1.10 (0.85, 1.43) | 1.11 (0.85, 1.44) | 1.09 (0.84, 1.42) |
| caesarean emergency | **1.26 (1.04, 1.51)*** | **1.24 (1.00, 1.50)*** | 1.19 (0.96, 1.48) | 1.19 (0.97, 1.48) | 1.17 (0.94, 1.46) |
| forceps or vacuum | 1.08 (0.86, 1.36) | 1.11 (0.86, 1.42) | 1.13 (0.86,1.46) | 1.13 (0.87, 1.46) | 1.13 (0.87, 1.46) |
| **Neonatal characteristics** | | | | | |
| **GA (weeks), N** | 14730/1340 | 11816/1078 | 11816/1078 | 11816/1078 | 11816/1078 |
| <37 | 1.19 (0.97, 1.47) | 1.18 (0.93, 1.50) | 1.17 (0.93, 1.49) | 1.14 (0.89, 1.45) | 1.14 (0.89, 1.45) |

*(Continued)*

**Table 4.** (Continued)

| Perinatal characteristics | Crude HR (95%CI) | Crude HR (95%CI) Complete data[a] | 1st Model Adj HR (95%CI) | 2nd Model Adj HR (95%CI) | 3rd Model Adj HR (95%CI) |
|---|---|---|---|---|---|
| 37–41 | Ref | Ref | Ref | Ref | Ref |
| ≥42 | 0.93 (0.75, 1.17) | 0.98 (0.77, 1.26) | 0.98 (0.77, 1.25) | 0.97 (0.76, 1.23) | 0.97 (0.76, 1.23) |
| **Birthweight for GA[c], N** | 14689/1332 | 11816/1078 | 11816/1078 | 11816/1078 | 11816/1078 |
| AGA | Ref | Ref | Ref | Ref | Ref |
| SGA | 1.04 (0.80, 1.36) | 0.97 (0.70, 1.33) | 0.97 (0.70, 1.34) | 0.95 (0.67, 1.30) | 0.93 (0.67, 1.28) |
| LGA | **1.30 (1.03, 1.64)*** | **1.37 (1.08, 1.76)*** | **1.32 (1.03, 1.70)*** | **1.32 (1.02, 1.68)*** | **1.31 (1.02, 1.68)*** |
| **Child infection-I[d], N** | 14179/1289 | 11361/1039 | 11361/1039 | 11361/1039 | 11361/1039 |
| No | Ref | Ref | Ref | Ref | Ref |
| Yes | 0.95 (0.70, 1.30) | 1.04 (0.75, 1.46) | 1.05 (0.74, 1.47) | 1.05 (0.75, 1.47) | 0.95 (0.49, 1.85) |
| **5-min Apgar, N** | 14676/1332 | 11773/1071 | 11773/1071 | 11773/1071 | 11773/1071 |
| ≥7 | Ref | Ref | Ref | Ref | Ref |
| <7 | **1.73 (1.16, 2.57)**** | **1.72 (1.16, 2.56)**** | **1.61 (1.02, 2.55)*** | 1.53 (0.97, 2.43) | 1.50 (0.94, 2.38) |
| **Neonatal care[e], N** | 11154/1014 | 9832/895 | 9832/895 | 9832/895 | 9832/895 |
| No | Ref | Ref | Ref | Ref | Ref |
| Yes | **1.40 (1.17, 1.66)***** | **1.30 (1.07, 1.57)**** | **1.28 (1.05, 1.56)*** | **1.24 (1.01, 1.52)*** | **1.25 (1.00, 1.56)*** |

N – number of total observations/number of events; GA – gestational age; IVF – in vitro fertilisation; BMI – body mass index; AGA – adequate for GA, SGA– small for GA, LGA – large for GA; NA – less than 10 observations;

*** p<0.001, ** p<0.01, * p<0.05;models 1–3: shaded are perinatal characteristics used as adjustment covariates in the respective model.

[a]according to complete data for all used adjustment covariates; [b]smoking and BMI at the time of enrolment into maternal health care; [c]calculated according to birthweight, sex and gestational age; [d]data according to the National Patient Register (in-patient care; since 1990); [e]data available since 1995

No associations were observed between early childhood infections and the risk of overall childhood cancer (Table 4) or specific cancer types (S2–S5 Tables).

### Associations between parental and pregnancy characteristics and risk of childhood cancer

High **maternal BMI** (overweight or obese) was associated with a higher risk of CNS tumors (adjusted HR 95%CI: 1.51, 1.04–2.21), lymphoma (adjusted HR 95%CI: 2.31 1.41–3.77), and somewhat increased risk of overall cancer (adjusted HR 95%CI: 1.11, 0.94–1.37; Tables 4 and 5), while low maternal BMI (underweight) was associated with a higher risk of leukaemia (adjusted HR 95%CI: 2.43, 1.40–4.22). Associations were independent of diagnosis with cancer predisposing syndromes (S6 Table), birthweight for GA, or gestational diabetes (S10 Table).

**Caesarean delivery** (CD), compared with spontaneous vaginal delivery, was associated with an increased risk of childhood cancer. Emergency CD was associated with an increased risk of overall childhood cancer (crude HR 95%CI: 1.26, 1.04–1.51), and both planned and emergency CD with increased risk of other cancer types combined (crude HR 95%CI: 1.45, 1.03–2.05 and 1.29, 1.01–1.75, respectively). After adjustments, only the association for planned CD and risk of other cancer types combined remained (adjusted HR 95%CI: 1.52 1.04–2.22) and was also independent of cancer cases diagnoses at birth (S1 Fig), at ≤6 months of age (Fig 2), diagnosis of cancer predisposing syndromes (S6 Table), and of birthweight for GA and Apgar score (S11 Table).

Furthermore, a **paternal age** of 35y or more compared to below 25y was associated with a reduced risk of lymphoma (adjusted HR 95%CI: 0.29, 0.12–0.71; Table 5). However, an analysis with a 5-year age advancement did not confirm these results (S12 Table).

**Table 5. Summarised associations of perinatal characteristics with the risk of leukaemia, CNS tumour, lymphoma and other cancer types combined (i.e., ICCC group IV-XII including unclassified diagnoses).**

| Perinatal characteristics | Crude HR (95%CI | Crude HR (95%CI) Complete data[a] | Model 1 HR (95%CI) | Model 2 HR (95%CI) | Model 3 HR (95%CI) |
|---|---|---|---|---|---|
| **LEUKAEMIA** | | | | | |
| *Parental characteristics* | | | | | |
| **Maternal BMI (kg/m²)[a], N** | 3268/ 300 | 3124/291 | 3124/291 | 3124/291 | 3124/291 |
| <18.5 | **2.34 (1.38, 3.97)\*\*** | **2.32 (1.34, 4.00)\*\*** | **2.44 (1.41, 4.23)\*\*** | **2.44 (1.41, 4.23)\*\*** | **2.43 (1.40, 4.22)\*\*** |
| 18.5–24.9 | Ref | Ref | Ref | Ref | Ref |
| 25–29.9 | 1.24 (0.96, 1.62) | 1.28 (0.98, 1.66) | 1.24 (0.94, 1.62) | 1.23 (0.94, 1.61) | 1.23 (0.94, 1.61) |
| ≥30 | 0.95 (0.64, 1.41) | 0.97 (0.65, 1.44) | 0.94 (0.63, 1.40) | 0.92 (0.62, 1.38) | 0.92 (0.62, 1.38) |
| *Neonatal characteristics* | | | | | |
| **Birthweight for GA, N** | 3782/343 | 3114/290 | 3114/290 | 3114/290 | 3114/290 |
| AGA | Ref | Ref | Ref | Ref | Ref |
| SGA | 1.07 (0.63, 1.79) | 0.97 (0.54, 1.73) | 0.96 (0.53, 1.72) | 0.93 (0.51, 1.68) | 0.94 (0.52, 1.72) |
| LGA | **1.66 (1.10, 2.51)\*** | **1.56 (1.01, 2.46)\*** | **1.58 (1.02, 2.51)\*** | **1.59 (1.01, 2.52)\*** | **1.58 (1.01, 2.51)\*** |
| **CNS TUMORS** | | | | | |
| *Parental characteristics* | | | | | |
| **Maternal BMI (kg/m²)[b], N** | 3027/275 | 2911/ 264 | 2911/ 264 | 2911/ 264 | 2911/ 264 |
| <18.5 | 0.86 (0.35, 2.09) | 0.94 (0.39, 2.29) | 0.99 (0.41, 2.43) | 0.99 (0.40, 2.42) | 0.99 (0.41, 2.43) |
| 18.5–24.9 | Ref | Ref | Ref | Ref | Ref |
| 25–29.9 | 1.09 (0.82, 1.45) | 1.15 (0.87, 1.53) | 1.19 (0.88, 1.57) | 1.18 (0.88, 1.57) | 1.17 (0.88, 1.57) |
| ≥30 | **1.31 (1.00, 1.99)\*** | **1.41 (1.01, 2.06)\*** | **1.51 (1.04, 2.21)\*** | **1.52 (1.04, 2.22)\*** | **1.51 (1.04, 2.21)\*** |
| **LYMPHOMA** | | | | | |
| *Parental characteristics* | | | | | |
| **Paternal age (years), N** | 1673/152 | 1322/115 | 1322/115 | 1322/115 | 1322/115 |
| <25 | Ref | Ref | Ref | Ref | Ref |
| 25-34 | 0.76 (0.44, 1.30) | 0.66 (0.36, 1.21) | **0.46 (0.21, 0.97)\*** | **0.44 (0.20, 0.94)\*** | **0.44 (0.20, 0.94)\*** |
| ≥35 | **0.54 (0.29, 0.99)\*** | 0.51 (0.26, 1.00) | **0.30 (0.13, 0.73)\*\*** | **0.29 (0.12, 0.71)\*\*** | **0.29 (0.12, 0.71)\*\*** |
| **Maternal BMI (kg/m²)[c], N** | 1372/124 | 1322/115 | 1322/115 | 1322/115 | 1322/115 |
| <18.5 | NA | NA | NA | NA | NA |
| 18.5–24.9 | Ref | Ref | Ref | Ref | Ref |
| 25–29.9 | 1.31 (0.87, 1.98) | 1.31 (0.87, 1.99) | 1.39 (0.91, 2.12) | 1.41 (0.91, 2.17) | 1.41 (0.91, 2.17) |
| ≥30 | **2.31 (1.41, 3.77)\*\*\*** | **2.10 (1.24, 3.54)\*\*\*** | **2.27 (1.33, 3.87)\*\*** | **2.26 (1.31, 3.87)\*\*** | **2.26 (1.31, 3.88)\*\*** |
| **OTHER CANCER TYPES COMBINED** | | | | | |
| *Pregnancy characteristics* | | | | | |
| **Mode of delivery, N** | 5687/517 | 4491/411 | 4491/411 | 4491/411 | 4491/411 |
| Vaginal no instruments | Ref | Ref | Ref | Ref | Ref |
| caesarean elective | **1.45 (1.03, 2.05)\*** | **1.55 (1.07, 2.26)\*** | **1.57 (1.08, 2.29)\*** | **1.57 (1.07, 2.29)\*** | **1.54 (1.06, 2.25)\*** |
| caesarean emergency | 1.29 (0.95, 1.75) | **1.39 (1.00, 1.94)\*** | **1.43 (1.02, 2.00)\*** | **1.45 (1.04, 2.04)\*** | 1.39 (0.98, 1.97) |
| forceps or vacuum | 0.98 (0.66, 1.48) | 1.06 (0.69, 1.63) | 1.11 (0.71, 1.72) | 1.11 (0.71, 1.73) | 1.11 (0.87, 1.82) |
| *Neonatal characteristics* | | | | | |
| **5-min Apgar, N** | 5666/514 | 4481/409 | 4481/409 | 4481/409 | 4481/409 |
| ≥7 | Ref | Ref | Ref | Ref | Ref |
| <7 | **2.39 (1.38, 4.14)\*\*** | **2.55 (1.36, 4.77)\*\*** | **2.60 (1.38, 4.91)\*\*** | **2.23 (1.17, 4.38)\*** | **2.16 (1.12, 4.15)\*** |

*(Continued)*

**Table 5.** (Continued)

| Perinatal characteristics | Crude HR (95%CI | Crude HR (95%CI) Complete data[a] | Model 1 HR (95%CI) | Model 2 HR (95%CI) | Model 3 HR (95%CI) |
|---|---|---|---|---|---|
| **Neonatal care[d], N** | 4103/373 | 3604/326 | 3604/326 | 3604/326 | 3604/326 |
| No | Ref | Ref | Ref | Ref | Ref |
| Yes | 1.49 (1.13, 2.00)** | 1.48 (1.10. 1.99)** | 1.50 (1.11, 2.02)** | 1.38 (1.01, 1.89)* | 1.40 (1.02, 1.97)* |

N, n of total observations/n of events;BMI- maternal body mass, AGA – adequate for GA, SGA – small for GA, LGA – large for GA; NA – less than 10 observations in either cases or controls; *** p<0.001, ** p<0.01 and * p<0.5); [a]according to complete data for all used adjustment covariates; [b]at enrolment into maternal health care, [c]calculated according to birthweight, sex, and gestational age; [d]data available since 1995.

Model 1: adjusted for maternal cancer, maternal and paternal age, maternal and paternal education, parity, maternal BMI and maternal smoking during early pregnancy.

Model 2: adjusted for variables in 1st model and additionally for IVF and mode of delivery.

Model 3: adjusted for all variables in 2nd model and additionally for gestational age;

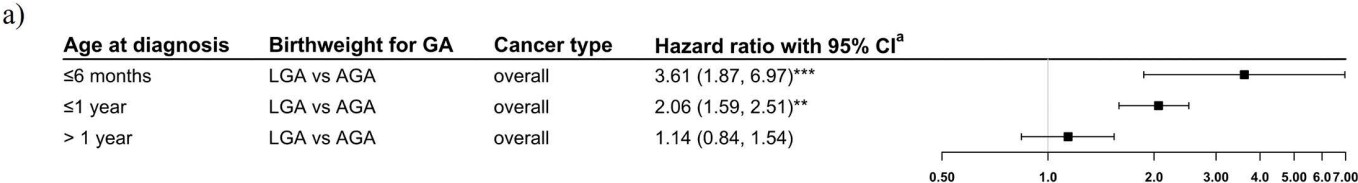

a)

| Age at diagnosis | Birthweight for GA | Cancer type | Hazard ratio with 95% CI[a] |
|---|---|---|---|
| ≤6 months | LGA vs AGA | overall | 3.61 (1.87, 6.97)*** |
| ≤1 year | LGA vs AGA | overall | 2.06 (1.59, 2.51)** |
| > 1 year | LGA vs AGA | overall | 1.14 (0.84, 1.54) |

b)

| Age at diagnosis | Mechanical ventilation | Cancer type | Hazard ratio with 95% CI[a] |
|---|---|---|---|
| ≤6 months | Yes vs No | overall | 2.54 (1.11, 5.80)* |
| | Yes vs No | other types combined | NA |
| ≤1 year | Yes vs No | overall | 2.19 (1.25, 3.84)** |
| | Yes vs No | other types combined | 2.12 (1.01, 4.46)* |
| > 1 year | Yes vs No | overall | 1.41 (0.93, 2.14) |
| | Yes vs No | other types combined | 1.90 (0.95, 3.83) |

c)

| Age at diagnosis | Birth delivery | Cancer type | Hazard ratio with 95% CI[a] |
|---|---|---|---|
| >6 months | Planed CD vs vaginal | overall | 1.10 (0.84, 1.45) |
| | Planed CD vs vaginal | other types combined | 1.63 (1.09, 2.44)* |
| 0-5 years | Planed CD vs vaginal | overall | 1.03 (0.72, 1.48) |
| | Planed CD vs vaginal | other types combined | 1.09 (0.63, 1.91) |
| 6-10 years | Planed CD vs vaginal | overall | 1.32 (0.74, 2.33) |
| | Planed CD vs vaginal | other types combined | NA |
| 11-18 years | Planed CD vs vaginal | overall | 0.87 (0.51, 1.50) |
| | Planed CD vs vaginal | other types combined | 1.77 (0.91, 3.39) |

**Fig 2. Associations of birthweight for GA (a), mechanical ventilation treatment (b) and planned caesarean delivery (c) with the risk of overall childhood cancer, and other cancer types combined, stratified by age at diagnosis.** (*** p<0.001, ** p<0.01, * p<0.05; a– adjusted according to model 3, NA – not available due to low observation number (i.e., <10); AGA – adequate for GA, SGA – small for GA, LGA – large for GA; other types combined – ICCC group IV-XII, including unclassified diagnoses).

We observed no associations between the risk of childhood cancer and other parental and pregnancy characteristics, except for children being diagnosed with cancer predisposing syndromes, which showed considerably higher risk of childhood cancer (HR 95%CI: 10.9, 7.79–15.5) (Tables 3 and S13). Sensitivity analyses conducted on participants with complete data showed no significant influence of missing data on the results (Tables 3 and 4).

## Discussion

We identified several perinatal characteristics independently associated with the risk of overall childhood cancer as well as specific cancer types. Most of the associations were observed for characteristics reflecting neonatal distress, including LGA, low 5-min Apgar score, and neonatal care treatment, which were associated with increased risk of overall childhood cancer, leukaemia, and/or other cancer types combined. Additionally, high maternal BMI was associated with an increased risk of lymphoma and CNS tumors, low maternal BMI with an increased risk of leukaemia, and planned caesarean delivery with an increased risk of other cancer types combined. While prior studies have reported associations between these characteristics and childhood cancer risk, none are yet considered established risk factors [4,10].

### Associations of neonatal characteristics and risk of childhood cancer

Several epidemiological studies have linked LGA with increased risk of leukaemia, lymphoma, CNS, kidney, liver, germ cell, and soft tissue tumours [10,31–35], while SGA has been linked with a higher risk of leukaemia, lymphoma, and CNS tumours [18,31,36]. In the present study, LGA demonstrated an increased risk of overall childhood cancer and leukaemia, and to some extent other cancer types combined, while no associations were observed between SGA and overall childhood cancer or cancer types. Foetal growth is a complex process influenced by the interplay of (epi)genetic, hormonal, and nutritional factors and the biological mechanisms linking foetal growth to the risk of childhood cancer remain unknown. Current principal hypotheses suggest that larger organs present more cells at risk of mutagenesis and that increased intrauterine cell proliferation may result from overexpression of growth factors like insulin-like growth factors I and II [10,18,31,33]. Moreover, maternal BMI and gestational diabetes were proposed as potential underlying factors of LGA-childhood cancer associations [37,38]. In this study, both BMI and gestational diabetes were associated with LGA (S14 Table), but none explained the associations between LGA and childhood cancer. Another hypothesis suggests the reverse effect of an *in-utero* present tumour on a foetus's growth [18]. Moreover, LGA has been highlighted as a risk factor for particularly early-diagnosed childhood cancer [31,33]. While exclusion of cases diagnosed at birth did not influence our results, we observed up to a three-fold increase in the risk of childhood cancer among children diagnosed ≤6 months. LGA could also result from certain underlying genetic syndromes (e.g., overgrowth Beckwith–Wiedemann syndrome) [39], which are well-known independent risk factors for certain childhood cancer types [39,40]. Nevertheless, the exclusion of children diagnosed with cancer predisposing syndromes (i.e., Down syndrome, Neurofibromatosis type 1, Congenital malformation syndromes involving early overgrowth, and Von Hippel-Lindau syndrome) did not affect observed associations with LGA.

A 5-min Apgar score <7 was associated with increased risk of all other cancers combined, however, its low prevalence (1.2%) prevented assessment of individual risks for CNS tumours, leukaemia, and lymphoma. Previous studies linked low 5-min Apgar scores with an increased risk of neuroblastoma, renal, CNS, and liver tumours [14,19,41–43] with studies on Swedish-Danish [14] and Brazilian cohorts [43] showing a stronger association among children diagnosed <6 months of age. Similarly, in our study, low 5-min Apgar scores were more frequent in children diagnosed ≤6 months than in those diagnosed later. This could imply a shared *in-utero* initiated aetiology of low Apgar score and childhood cancer [14]. As shown in this study, low Apgar scores often co-occur with SGA and neonatal treatments [14], both linked to higher childhood cancer risk [36,44,45]. However, in our study, these factors do not explain the observed association between Apgar scores and the increased risk of other cancers combined.

Various studies have suggested long-term adverse health effects of neonatal treatments, including risk for malignancy [19,46–48]. A Finnish study reported an association between mechanical ventilation, resuscitation, and antibiotic therapy,

with risk of overall childhood cancer, irrespective of GA [19], while the increased risk of childhood cancer following oxygen supplementation was also reported by others [45,49]. In this study, overall neonatal treatments were associated with an increased risk of other cancer types combined. However, this association disappeared after excluding children diagnosed with cancer predisposing syndromes. Low statistical power hampered our investigation of the specific above-mentioned neonatal treatments, except for mechanical ventilation, which indicated a higher risk of overall childhood cancer and other cancer types combined. Mechanical ventilation follows a diagnosis of neonatal hypoxia and asphyxia, which often co-occur with cancer risk factors such as low Apgar score and high birthweight [19,49]. In this study, the association between mechanical ventilation and childhood cancer risk was independent of those factors as well as of the diagnosis of cancer predisposing syndromes. Oxygen supplementation can generate reactive oxygen species, implicated in carcinogenesis and cancer progression, though evidence remains conflicting [50]. As concluded by other studies, observed associations do not prove causation, and oxygen supplementation should be considered only as potentially carcinogenic [19,49]. Neo-natal intubation is commonly accompanied by diagnostic radiation, another possible carcinogenic risk factor [4,19]. Other unmeasured factors—such as pregnancy complications, maternal health conditions, and congenital malformations—may predispose infants to both neonatal therapies and childhood cancer, suggesting possible reverse causality. Thus, the link between neonatal therapies and childhood cancer warrants further study.

## Parental and pregnancy characteristics and risk of childhood cancer

Maternal overweight or obesity was previously associated with an increased risk of childhood cancer, including leukae-mia, CNS tumours, and retinoblastoma [11,51–53], while others reported null associations [11,31,54,55]. In our study, we observed an association between maternal obesity in early pregnancy and a higher risk of CNS tumours and lymphoma as well as a trend of maternal overweight being associated with increased risk of leukaemia. Those associations were independent of gestational diabetes or birthweight for GA; characteristics commonly associated with maternal BMI and childhood cancer [37]. Additionally, we observed a higher risk of leukaemia among children born to underweight moth-ers. Maternal underweight has been linked to an increased risk of germ cell tumours and overall cancer in some studies [51,54], whereas others have reported null associations [11]. The inconsistent results across studies could result from variations in study designs or methods in assessing BMI as a risk factor (e.g., continuous, 5-unit change, or categories).

The mechanisms linking maternal malnutrition to childhood cancer remain unclear, though altered foetal or neonatal epigenetic programming is the leading hypothesis [37]. Maternal obesity has been associated with differential DNA meth-ylation in newborns, including cancer-related genes such as *IGF1*, *IGF2*, and *MAD1L1* [56–58]. A recent longitudinal study found persistent methylation changes linked to metabolic and developmental pathways through the first year of life [59], suggesting lasting epigenetic effects. Similarly, offspring of underweight women may experience epigenetic dysregulation in growth and metabolism genes [60], though evidence is limited and largely from low-income settings. With maternal overweight and obesity rising globally, from 25% to 46% in Sweden between 1992 and 2022 [61], improving maternal nutrition before and during pregnancy could be an important cancer prevention strategy.

Moreover, we observed caesarean delivery to be associated with an increased risk of overall childhood cancer. The association was observed particularly for planned CD and risk of other cancer types combined, which was independent of additional adjustment for 5-min Apgar score or birthweight for GA. Globally, the caesarean delivery rate is increasing [62] and so is the number of studies reporting its association with the risk of childhood cancer, particularly for leukae-mia, but also CNS, liver, and other solid tumours [12,63–66]. While a large cohort with over 11 thousand across Swe-den, Denmark and Finland reported null associations [67]. However, in line with our results, several studies reported planned rather than emergency CD, as significantly associated with childhood cancer [63–66,68]. Commonly suggested mechanisms are limited exposure to maternal vaginal microflora and lower stress hormone levels (e.g., cortisol, adren-aline) in neonates born via planned CD, compared to those born via emergency CD or vaginal delivery [64,66,69]. Both low microbial diversity as well as hypocortisolism result in a weakened neonatal immune system [70], which

might consequently contribute to a higher risk of cancer development [64,66,71]. Marcoux et al., 2023 [64] suggested that including early-detected cases may have biased prior CD associations through perinatal confounding. In contrast, excluding early-onset cases in our study strengthened the association between planned CD and risk of other combined cancer types. This discrepancy may reflect CD type, as Marcoux et al. (2022) did not distinguish emergency from planned CD, which represent different exposure scenarios [72]. In summary, emerging evidence and plausible mechanisms linking planned CD to childhood cancer highlight the need to reduce non-medically indicated CDs, commonly referred to as "caesarean on maternal request".

Parental age and childhood cancer risk have been widely studied, with mixed findings ranging from increased risk with advanced parental age [16,72–75] to no associations [12,76,77]. In the present study, higher paternal age (25–34 and ≥35 vs. <25) was linked to lower lymphoma risk, but this was not confirmed using 5-year age increments. Prior evidence on paternal age and lymphoma is also inconsistent [16,73,78]; the observed protective role of paternal age on the risk of lymphoma in our study should be interpreted with caution.

## Strength and limitations of the study

The study's strengths include the use of several high-quality health- and population-based registries, enabling the investigation of several perinatal factors and adjustment for relevant confounders. Additionally, objectively and prospectively collected registry data eliminated the risk of selection and recall bias. Moreover, the availability of data for some genetic syndromes (i.e., Down syndrome, Neurofibromatosis type 1, Congenital malformation syndromes involving early overgrowth (Beckwith-Wiedemann syndrome), and Von Hippel-Lindau syndrome) enabled us to make additional adjustments for these known cancer-predisposing syndromes. The main limitation of the FeToxCancer study is a low number of cases per specific cancer types, which resulted in low statistical power and hindered a more comprehensive analysis of perinatal characteristics with the risk of specific cancer types. The statistical analysis of this study includes multiple comparisons, increasing the risk of false positives. However, given the limited statistical power, lowering the significance threshold could potentially overlook meaningful associations. Therefore, we chose to interpret the results cautiously in the context of existing literature rather than adjust for multiple comparisons. Nevertheless, the possibility of false positives still exists. Preterm birth reflects both immaturity and various pathological factors, with currently unclear links to GA. Consequently, in studies on neonatal outcomes, adjusting for GA, as a potential mediator factor, may introduce collider bias [30]. In our study, we examined numerous maternal and pregnancy characteristics, including GA, as potential risk factors for childhood cancer, many of which have unclear relationships with GA. To assess GA both as a potential risk factor and a confounder (e.g., neonatal care, mechanical ventilation therapy) while addressing collider bias, we fitted models with and without GA, as previously proposed by Wilcox et al. (2011) [30]. Lastly, missing data for some characteristics (e.g., BMI), as well as a lack of information on other risk factors such as family history, and environmental exposures, might represent bias in the present study.

## Conclusions

The present study provides support for several perinatal characteristics previously associated with the risk of childhood cancer. The reported results strongly highlight the significance of the perinatal period as a critical window, particularly for early-onset childhood cancer. Furthermore, our study contributes additional insight into risk prediction as well as provides knowledge for evidence-based care of mothers and newborns, which may facilitate the improvement of preventive measures.

## Supporting information

**S1 Table. Distribution of studied perinatal characteristics between cases of leukaemia, CNS tumours, lymphoma and other cancer types combined and matching controls.**
(DOCX)

**S2 Table. Associations of perinatal characteristics with leukaemia.**
(DOCX)

**S3 Table. Associations of perinatal characteristics with CNS tumours.**
(DOCX)

**S4 Table. Associations of perinatal characteristics with lymphoma.**
(DOCX)

**S5 Table. Associations of perinatal characteristics with other cancer types combined (ICCC groups IV to XII including by ICCC unclassified).**
(DOCX)

**S6 Table. Associations of perinatal characteristics with risk of childhood cancer and cancer types after exclusion of children diagnosed with cancer-predisposing syndromes[#].**
(DOCX)

**S7 Table. Association of birthweight for GA with risk of overall childhood cancer and leukaemia after additional adjustment for gestational diabetes.**
(DOCX)

**S8 Table. Association of 5-min Apgar with other cancer types combined after additional adjustment for birthweight for GA and neonatal care (a) and distribution of 5-min Apgar according to the age at diagnosis (b).**
(DOCX)

**S9 Table. Distribution and association of specific neonatal therapies with overall childhood cancer (a), and association of mechanical ventilation with other cancer types (b).**
(DOCX)

**S10 Table. Association of maternal BMI with leukemia, CNS tumor and lymphoma after additional adjustment for birthweight for GA (a) or gestational diabetes (b).**
(DOCX)

**S11 Table. Association between mode of delivery and risk of other cancer types combined after additional adjustment for birthweight for GA and 5-min Apgar score.**
(DOCX)

**S12 Table. Association between parental 5-year advance in age with risk of childhood cancer and specific cancer types.**
(DOCX)

**S13 Table. Distribution and associations of additionally tested perinatal characteristics with childhood cancer.**
(DOCX)

**S14 Table. Associations of birthweight for GA with maternal BMI and Gestational diabetes (a) and of 5-min Apgar with birthweight for GA, gestational age and admission to neonatal care (b).**
(DOCX)

**S1 Fig. Forest plot presenting associations of birthweight for GA, 5-min Apgar score, admission to neonatal care, mechanical ventilation treatment, and mode of delivery with the risk of overall childhood cancer, leukaemia or other cancer types combined after the exclusion of cancer cases diagnosed at birth (adjusted according to model**

3; ** p<0.01, *p<0.05; other types combined– ICCC groups IV-XII including unclassified diagnoses; AGA – adequate for GA, SGA–small for GA, LGA–large for GA).
(DOCX)

## Author contributions

**Conceptualization:** Anja Stajnko, Anna Oudin, Christian Lindh, Ingrid Øra, Jenny Selander, Lars Rylander, Maria Albin, Karin Källén, Karin Broberg.

**Formal analysis:** Anja Stajnko.

**Funding acquisition:** Karin Broberg.

**Investigation:** Anja Stajnko, Karin Broberg.

**Project administration:** Karin Broberg.

**Resources:** Karin Broberg.

**Supervision:** Karin Källén, Karin Broberg.

**Visualization:** Anja Stajnko.

**Writing – original draft:** Anja Stajnko.

**Writing – review & editing:** Anja Stajnko, Jesse Daniel Thacher, Anna Oudin, Christian Lindh, Thomas Lundh, Ingrid Øra, Jenny Selander, Lars Rylander, Maria Albin, Karin Källén, Karin Broberg.

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
