## [Decision Letter · Decision Letter 0]

14 Oct 2025

Dear Dr. Broberg,

Thank you for submitting your manuscript to PLOS ONE. After careful consideration, we feel that it has merit but does not fully meet PLOS ONE’s publication criteria as it currently stands. Therefore, we invite you to submit a revised version of the manuscript that addresses the points raised during the review process.

A rebuttal letter that responds to each point raised by the academic editor and reviewer(s). You should upload this letter as a separate file labeled ‘Response to Reviewers’.A marked-up copy of your manuscript that highlights changes made to the original version. You should upload this as a separate file labeled ‘Revised Manuscript with Track Changes’.An unmarked version of your revised paper without tracked changes. You should upload this as a separate file labeled ‘Manuscript’.

We look forward to receiving your revised manuscript.

Kind regards,

Quetzal A. Class, PhD

Academic Editor

PLOS ONE

**Journal Requirements:**

1. When submitting your revision, we need you to address these additional requirements. Please ensure that your manuscript meets PLOS ONE’s style requirements, including those for file naming. The PLOS ONE style templates can be found at https://journals.plos.org/plosone/s/file?id=wjVg/PLOSOne_formatting_sample_main_body.pdf and https://journals.plos.org/plosone/s/file?id=ba62/PLOSOne_formatting_sample_title_authors_affiliations.pdf 2. Thank you for stating the following financial disclosure: a) -KB,  -Grant No: Dnr 20 0825 Pj; -full name: Swedish Cancer Society, -URL: https://www.cancerfonden.se/om-oss/about-nob) -KB; -Grant No: PR2020-0012; -full name: Swedish Childhood Cancer Foundation; -URL: https://www.barncancerfonden.se;-noc) KB; -Grant No: 2022-Projekt0162; -Full name: ALF; -url: https://www.intramed.lu.se/en/research/alf-funding;-nod) -KB, -Grant No: 2022-01-11:10, -full name: Sjöberg Foundation, -URL: https://sjobergstiftelsen.se/-no  Please state what role the funders took in the study.  If the funders had no role, please state: “The funders had no role in study design, data collection and analysis, decision to publish, or preparation of the manuscript.” If this statement is not correct you must amend it as needed. Please include this amended Role of Funder statement in your cover letter; we will change the online submission form on your behalf. 3. We note that you have indicated that there are restrictions to data sharing for this study. For studies involving human research participant data or other sensitive data, we encourage authors to share de-identified or anonymized data. However, when data cannot be publicly shared for ethical reasons, we allow authors to make their data sets available upon request. For information on unacceptable data access restrictions, please see http://journals.plos.org/plosone/s/data-availability#loc-unacceptable-data-access-restrictions.  Before we proceed with your manuscript, please address the following prompts: a) If there are ethical or legal restrictions on sharing a de-identified data set, please explain them in detail (e.g., data contain potentially identifying or sensitive patient information, data are owned by a third-party organization, etc.) and who has imposed them (e.g., a Research Ethics Committee or Institutional Review Board, etc.). Please also provide contact information for a data access committee, ethics committee, or other institutional body to which data requests may be sent. b) If there are no restrictions, please upload the minimal anonymized data set necessary to replicate your study findings to a stable, public repository and provide us with the relevant URLs, DOIs, or accession numbers. Please see http://www.bmj.com/content/340/bmj.c181.long for guidelines on how to de-identify and prepare clinical data for publication. For a list of recommended repositories, please see https://journals.plos.org/plosone/s/recommended-repositories. You also have the option of uploading the data as Supporting Information files, but we would recommend depositing data directly to a data repository if possible. Please update your Data Availability statement in the submission form accordingly. 4. We note that you have included the phrase “data not shown” in your manuscript. Unfortunately, this does not meet our data sharing requirements. PLOS does not permit references to inaccessible data. We require that authors provide all relevant data within the paper, Supporting Information files, or in an acceptable, public repository. Please add a citation to support this phrase or upload the data that corresponds with these findings to a stable repository (such as Figshare or Dryad) and provide and URLs, DOIs, or accession numbers that may be used to access these data. Or, if the data are not a core part of the research being presented in your study, we ask that you remove the phrase that refers to these data. 5. We note that Figure 1 in your submission contain map images which may be copyrighted. All PLOS content is published under the Creative Commons Attribution License (CC BY 4.0), which means that the manuscript, images, and Supporting Information files will be freely available online, and any third party is permitted to access, download, copy, distribute, and use these materials in any way, even commercially, with proper attribution. For these reasons, we cannot publish previously copyrighted maps or satellite images created using proprietary data, such as Google software (Google Maps, Street View, and Earth). For more information, see our copyright guidelines: http://journals.plos.org/plosone/s/licenses-and-copyright. We require you to either present written permission from the copyright holder to publish these figures specifically under the CC BY 4.0 license, or remove the figures from your submission: a. You may seek permission from the original copyright holder of Figure 1 to publish the content specifically under the CC BY 4.0 license.   We recommend that you contact the original copyright holder with the Content Permission Form (http://journals.plos.org/plosone/s/file?id=7c09/content-permission-form.pdf) and the following text:“I request permission for the open-access journal PLOS ONE to publish XXX under the Creative Commons Attribution License (CCAL) CC BY 4.0 (http://creativecommons.org/licenses/by/4.0/). Please be aware that this license allows unrestricted use and distribution, even commercially, by third parties. Please reply and provide explicit written permission to publish XXX under a CC BY license and complete the attached form.” Please upload the completed Content Permission Form or other proof of granted permissions as an “Other” file with your submission. In the figure caption of the copyrighted figure, please include the following text: “Reprinted from [ref] under a CC BY license, with permission from [name of publisher], original copyright [original copyright year].” b. If you are unable to obtain permission from the original copyright holder to publish these figures under the CC BY 4.0 license or if the copyright holder’s requirements are incompatible with the CC BY 4.0 license, please either i) remove the figure or ii) supply a replacement figure that complies with the CC BY 4.0 license. Please check copyright information on all replacement figures and update the figure caption with source information. If applicable, please specify in the figure caption text when a figure is similar but not identical to the original image and is therefore for illustrative purposes only.The following resources for replacing copyrighted map figures may be helpful: USGS National Map Viewer (public domain): http://viewer.nationalmap.gov/viewer/The Gateway to Astronaut Photography of Earth (public domain): http://eol.jsc.nasa.gov/sseop/clickmap/Maps at the CIA (public domain): https://www.cia.gov/library/publications/the-world-factbook/index.html and https://www.cia.gov/library/publications/cia-maps-publications/index.htmlNASA Earth Observatory (public domain): http://earthobservatory.nasa.gov/Landsat:
http://landsat.visibleearth.nasa.gov/USGS EROS (Earth Resources Observatory and Science (EROS) Center) (public domain): http://eros.usgs.gov/#Natural Earth (public domain): http://www.naturalearthdata.com/ 6. Please upload a new copy of Figure 2 as the detail is not clear. Please follow the link for more information:  https://journals.plos.org/plosone/s/figures 7. Please include captions for your Supporting Information files at the end of your manuscript, and update any in-text citations to match accordingly. Please see our Supporting Information guidelines for more information: http://journals.plos.org/plosone/s/supporting-information. 8. If the reviewer comments include a recommendation to cite specific previously published works, please review and evaluate these publications to determine whether they are relevant and should be cited. There is no requirement to cite these works unless the editor has indicated otherwise. 

Reviewers’ comments:

**Comments to the Author**

1. Is the manuscript technically sound, and do the data support the conclusions?

Reviewer #1: Yes

Reviewer #2: Yes

Reviewer #3: Partly

2. Has the statistical analysis been performed appropriately and rigorously?

Reviewer #1: Yes

Reviewer #2: No

Reviewer #3: No

3. Have the authors made all data underlying the findings in their manuscript fully available?

Reviewer #1: Yes

Reviewer #2: No

Reviewer #3: Yes

4. Is the manuscript presented in an intelligible fashion and written in standard English?

Reviewer #1: Yes

Reviewer #2: Yes

Reviewer #3: Yes

**Reviewer #1:** The manuscript presents a large-scale population-based case-control study investigating associations between various perinatal factors and childhood cancer risk. The topic is timely, highly relevant, and aligns with PLOS ONE’s focus on scientifically rigorous, ethically conducted research that contributes meaningfully to its field.The manuscript presents a large-scale population-based case-control study investigating associations between various perinatal factors and childhood cancer risk. The topic is timely, highly relevant, and aligns with PLOS ONE’s focus on scientifically rigorous, ethically conducted research that contributes meaningfully to its field.The manuscript presents a large-scale population-based case-control study investigating associations between various perinatal factors and childhood cancer risk. The topic is timely, highly relevant, and aligns with PLOS ONE’s focus on scientifically rigorous, ethically conducted research that contributes meaningfully to its field.The manuscript presents a large-scale population-based case-control study investigating associations between various perinatal factors and childhood cancer risk. The topic is timely, highly relevant, and aligns with PLOS ONE’s focus on scientifically rigorous, ethically conducted research that contributes meaningfully to its field.

Strengths:

The study is based on a well-powered sample (1340 cases, 13,400 controls).

Comprehensive use of Swedish national registries ensures high data reliability.

The use of Cox regression with age as the underlying time scale is a strength over simple logistic regression.

Results are adjusted in a stepwise fashion, providing robustness.

Weaknesses / Suggestions:

Collider bias is acknowledged, and the strategy of multiple models (with/without GA) is commendable. Still, a more detailed justification for the modeling strategy should be included in the Discussion.

Multiple comparisons: The manuscript reports a large number of statistical tests. While correction was intentionally not applied, this choice should be more clearly defended in the limitations section.

Covariate selection: Certain known risk factors (e.g., environmental exposures, family history) are not accounted for. This should be addressed as a limitation.

IVF data: The availability from 1989-2006 and 2007-2021 in separate registries should be more clearly discussed in terms of its impact on data completeness.

Clarity of Writing

Strengths:

The manuscript is overall clearly structured and easy to follow. The English is fluent and mostly well-polished.

Suggestions:

Some paragraphs in the Discussion are overly long and could be made more concise.

Avoid repetition when reporting results, especially when hazard ratios are reiterated in both Results and Discussion.

Minor language edits are needed: Consider proofreading by a native speaker or editor for minor grammatical refinements.

Recommendation: Minor Revision

The manuscript is scientifically sound and of high relevance. The authors should address the following in their revision:

Provide a more explicit discussion on the lack of multiple testing correction.

Briefly elaborate on the implications of the limited IVF data period.

Clarify limitations such as residual confounding and registry-based constraints.

Undertake light English proofreading to enhance polish.

**Reviewer #2:** In this manuscript the authors have examined the association between several perinatal characteristics and the occurrence of childhood cancer using a register-based case-control data that includes all children diagnosed with cancer in southern Sweden between 1989-2021. The authors found several statistically significant associations: while these associations have been reported in previous studies, results were often conflicting. My main comment to the authors is that I believe that they should have used a conditional logistic regression model to analyze their data instead of a Cox proportional hazards model. See below for more details on this as well as additional comments on the manuscript. In this manuscript the authors have examined the association between several perinatal characteristics and the occurrence of childhood cancer using a register-based case-control data that includes all children diagnosed with cancer in southern Sweden between 1989-2021. The authors found several statistically significant associations: while these associations have been reported in previous studies, results were often conflicting. My main comment to the authors is that I believe that they should have used a conditional logistic regression model to analyze their data instead of a Cox proportional hazards model. See below for more details on this as well as additional comments on the manuscript. In this manuscript the authors have examined the association between several perinatal characteristics and the occurrence of childhood cancer using a register-based case-control data that includes all children diagnosed with cancer in southern Sweden between 1989-2021. The authors found several statistically significant associations: while these associations have been reported in previous studies, results were often conflicting. My main comment to the authors is that I believe that they should have used a conditional logistic regression model to analyze their data instead of a Cox proportional hazards model. See below for more details on this as well as additional comments on the manuscript. In this manuscript the authors have examined the association between several perinatal characteristics and the occurrence of childhood cancer using a register-based case-control data that includes all children diagnosed with cancer in southern Sweden between 1989-2021. The authors found several statistically significant associations: while these associations have been reported in previous studies, results were often conflicting. My main comment to the authors is that I believe that they should have used a conditional logistic regression model to analyze their data instead of a Cox proportional hazards model. See below for more details on this as well as additional comments on the manuscript.

- Abstract: please indicate that the numbers in parentheses represent hazard ratios and 95% confidence intervals.

- Abstract: add to the abstract that the 1340 cancer cases were born in southern Sweden between 1989-2021

- Regarding controls selection, the authors wrote “The inclusion criteria were being born between 1989 and 2021 with no prior history of cancer diagnosis or recorded death before the age of 19 while allowing for cancer diagnosis at the age of 19 or older.” I guess that also controls did not emigrate before age 19, or otherwise you wouldn’t know whether they died or got a cancer diagnosis. However, since the controls cannot die before age 19 and some of the assessed perinatal characteristics are linked to childhood mortality (especially infant mortality) I wonder whether the decision to select controls only among children who didn’t die before age 19 could have led to a slight overestimation of the risk estimates, as these controls likely had more normal perinatal characteristics than the general population (that includes children who died during childhood). Moreover, I think that this sentence should be rephrased since it is not possible to check whether potential controls born in the more recent years are cancer free at age 19 (or have not died by age 19) since they have not reached this age during the study period

- Small for gestational age is defined as a birth weight below the 10th percentile for a specific gestational age, and these percentiles are usually estimated separately for boys and girls. This means that, in the general population, there are approximately 10% of children that are born SGA. However, when defining SGA as SD < -2, there will be a bit more than 2-3% of children defined as being born SGA (in Table 1 the authors reported that it was 4%, which is much lower than the definition of SGA). The same reasoning applies to LGA. Can the authors explain why they have defined SGA/LGA in a different way?

- Information regarding child infections was obtained from the National Patient Register. Why did the authors use only the inpatient register and did not use the outpatient register data (that started in 2001)? Moreover, the authors should write in the text that they use inpatient care data, this is reported only in Table 1

- The authors wrote that “Traditional case-control analyses using logistic regression do not allow estimation of time-varying associations” and therefore used Cox regression models to analyze their matched case-control data. However, I disagree with them because in a case-control study you can assess time varying associations, for example the effect of an exposure on childhood cancer risk during the first five years of life, by restricting the analyses to the cases diagnosed during the age period of interest and their matched controls. It is with time-varying covariates/exposures that one should use Cox regression when analyzing case-control data. I am not an expert on using Cox regression on case-control data but from the study below from Statistics in Medicine it seems that the naïve approach (analyzing the case-control data using the standard Cox regression model as if the data were obtained from a cohort study) could lead to an underestimation of the association. I therefore suggest to the authors to re-run the analyses using a conditional logistic regression model, as all the exposures that they have analyzed (perinatal factors) are not time varying.

Reference: Statistics in Medicine (2003) - Evaluation of Cox’s model and logistic regression for matched case-control data with time-dependent covariates: a simulation study

- The authors wrote “The models were not assessed for characteristics or categories with fewer than ten observations.” Please clarify whether with “ten observations” you mean ten cancer cases for a specific category or if with ten observations you meant a combination of cases and controls

- Why did the authors add a quadratic term for BMI in the regression models? In the main analysis BMI was used as a categorical variable rather than a continuous one

- Did the authors have information regarding genetic syndromes? As these are linked to childhood cancer risk and could also be linked to some perinatal characteristics, is it possible to additionally adjust for genetic syndromes, at least for the most common ones such as Down syndrome, neurofibromatosis type 1, etc.?

- In the discussion the authors wrote “However, several studies, including a large cohort with over 11 thousand cases from Sweden, Denmark and Finland, reported null associations (65,66).”. However, references 65 and 66 refer to studies performed in the UK and the US. Please check that all references are cited correctly.

- I agree with the authors’ decision to not adjust for multiple comparisons. However, the authors should mention in the discussion section that because of the low number of cases and the several associations analyzed, it is possible that some of the reported statistically significant associations could be chance findings.

- In the last paragraph of the discussion, the authors wrote that “Nevertheless, as genetic predisposition explains less than 10% of childhood cancer cases (4), we believe that other perinatal factors, such as environmental factors, could contribute to our findings.” The authors also stated in the conclusion paragraph that “The reported results strongly highlight the significance of the perinatal period as a critical window, particularly for early-onset childhood cancer”. Can the authors report and comment on the proportion of childhood cancer cases that are caused by the exposure assessed in the current study? Moreover, in the abstract the authors wrote “Results indicate several perinatal characteristics associated with the risk of childhood cancer, which is important for prevention”. If the authors believe that these results are important for prevention, this should be thoroughly discussed in the discussion section, together with a discussion on the potential number of cases explained by these exposures and how these results can be used for preventive measures.

Minor comments:

- Check the spelling used throughout the manuscript as sometimes you write leukemia and other times you write leukaemia

- Table 2: Why some ICCC-3 groups are written in italic?

- Table 2: the columns percentages in some instances are not correct and when summed together gave a total > 100%

- Tables should be presented in the text in numerical order. In the text, Table 3 is mentioned before Table 2

- Footnote Table 4: did the authors mean to write inpatient register rather than incoming patient register

- Table 4: why are some numbers underlined in table 4?

**Reviewer #3:** This is a reasonable study design given the available sample. The major variables of perinatal characteristics, parental education and GA are well formulated as necessary. The variable exclusions for missingness, incompleteness or otherwise made sense.This is a reasonable study design given the available sample. The major variables of perinatal characteristics, parental education and GA are well formulated as necessary. The variable exclusions for missingness, incompleteness or otherwise made sense.This is a reasonable study design given the available sample. The major variables of perinatal characteristics, parental education and GA are well formulated as necessary. The variable exclusions for missingness, incompleteness or otherwise made sense.This is a reasonable study design given the available sample. The major variables of perinatal characteristics, parental education and GA are well formulated as necessary. The variable exclusions for missingness, incompleteness or otherwise made sense.

The statistical analysis section starts on line 165. The investigators give a logical rationale for the approach stating that, ‘Traditional case-control analyses using logistic regression do not allow estimation of time-varying associations. Therefore, in the present study, the Cox proportional hazards regression with child age as the underlying time scale was used to examine the association between perinatal characteristics (Table 1) and risk of childhood cancer.’ Their checking for the appropriateness of proportional hazards was followed appropriately and thus the major model for presenting the results as hazard ratios and their interpretation was reasonable in a somewhat time dependency situation.

However, their explanation of the GA as a potential risk factor of childhood cancer and for most of these characteristics the relationships with GA still remains unclear in this context from the statistical perspective. They note that to investigate GA as a potential risk factor while addressing potential collider bias, they conducted three separate models as outlined. Some detail about the process with actual variable examples would be helpful to the reader.

The Results start on line 207 and series of descriptive tables follow. The tables have so much detail that they are cumbersome to get through especially Tables S1 to S13. There is no discussion of any confounding influences (since adjustments were made as per the investigators), just a summary list of possibly associative characteristics with the outcome. Where there any competing risks in the process?

The limitations are noted on page 17. The investigators note the low numbers in some categories resulting in low power. Was the geographic coverage too small or initial sampling period too narrow. Where does power come into this situation which is a case control study? How relevant is the county of birth in the matching process or is this tied to a genetic or environmental consideration?

.

Reviewer #1: **Yes:** Nihat Bugra AgaogluNihat Bugra AgaogluNihat Bugra AgaogluNihat Bugra Agaoglu

Reviewer #2: No

Reviewer #3: No

---

## [Author Response · Author response to Decision Letter 1]

22 Dec 2025

Letter of response

We would like to thank the editor and the reviewers for their time reviewing the manuscript and for their constructive comments and suggestions, which have improved the quality of the manuscript. We have addressed all comments and revised the manuscript accordingly. Our point-by-point responses are given below.

Reviewer #1:

The manuscript presents a large-scale population-based case-control study investigating associations between various perinatal factors and childhood cancer risk. The topic is timely, highly relevant, and aligns with PLOS ONE’s focus on scientifically rigorous, ethically conducted research that contributes meaningfully to its field.

Strengths:The study is based on a well-powered sample (1340 cases, 13,400 controls). Comprehensive use of Swedish national registries ensures high data reliability. The use of Cox regression with age as the underlying time scale is a strength over simple logistic regression. Results are adjusted in a stepwise fashion, providing robustness.

Weaknesses / Suggestions:

-Collider bias is acknowledged, and the strategy of multiple models (with/without GA) is commendable. Still, a more detailed justification for the modelling strategy should be included in the Discussion.

As suggested, we have included the following sentence in the discussion of limitations of the study: “Preterm birth reflects both immaturity and various pathological factors, with currently unclear links to GA. Consequently, in studies on neonatal outcomes, adjusting for GA, as a potential mediator factor, may introduce collider bias (27). In our study, we examined numerous maternal and pregnancy characteristics, including GA, as potential risk factors for childhood cancer, many of which have unclear relationships with GA. To assess GA as both a potential risk factor and a true confounder (e.g. neonatal care, mechanical ventilation therapy) while addressing collider bias, we fitted models both with and without GA, as previously proposed by Wilcox et al., 2011 (27).”

-Multiple comparisons: The manuscript reports a large number of statistical tests. While correction was intentionally not applied, this choice should be more clearly defended in the limitations section.

According to comments and suggestions of reviewers 1 and 2, the following was added to the discussion of limitations of the study: “The statistical analysis includes multiple comparisons, increasing the risk of false positives; however, given the limited statistical power, lowering the significance threshold could potentially overlook biologically meaningful associations. Therefore, we chose to interpret the results cautiously in the context of existing literature rather than adjust for multiple comparisons, still acknowledging that some statistically significant associations may be due to chance. “

-Covariate selection: Certain known risk factors (e.g., environmental exposures, family history) are not accounted for. This should be addressed as a limitation.

This information is included in the conclusion: “Moreover, missing data for some characteristics (e.g., BMI), as well as lack of information on risk factors such as family history and environmental exposures, might represent bias in the current study.”

-IVF data: The availability from 1989-2006 and 2007-2021 in separate registries should be more clearly discussed in terms of its impact on data completeness.

In this study, we have investigated whether assisted pregnancy with IVF (yes vs. no) was associated with the risk of childhood cancer. The Swedish National Quality Register of Assisted Reproduction (Q-IVF) was initiated in 2007; however, for the study period preceding the year of 2007, we could access the same information from the National Medical Birth Register for all study participants. As such, the inclusion of two separate registries to obtain this information did not impact the completeness of our data (i.e. information is available for the whole study population; see Table 2).

Clarity of Writing

Strengths:The manuscript is overall clearly structured and easy to follow. The English is fluent and mostly well-polished.

Suggestions:

-Some paragraphs in the Discussion are overly long and could be made more concise.

As suggested, we have made several discussion paragraphs more concise.

-Avoid repetition when reporting results, especially when hazard ratios are reiterated in both Results and Discussion.

We have removed the repetitions of the results within the discussion.

-Minor language edits are needed: Consider proofreading by a native speaker or editor for minor grammatical refinements.

As suggested, the manuscript was proofread by a native speaker.

Reviewer #2: In this manuscript the authors have examined the association between several perinatal characteristics and the occurrence of childhood cancer using a register-based case-control data that includes all children diagnosed with cancer in southern Sweden between 1989-2021. The authors found several statistically significant associations: while these associations have been reported in previous studies, results were often conflicting. My main comment to the authors is that I believe that they should have used a conditional logistic regression model to analyze their data instead of a Cox proportional hazards model. See below for more details on this as well as additional comments on the manuscript.

- Abstract: please indicate that the numbers in parentheses represent hazard ratios and 95% confidence intervals.

As suggested, we have added HR, 95CI% into all parentheses.

- Abstract: add to the abstract that the 1340 cancer cases were born in southern Sweden between 1989-2021

Revised as suggested.

- Regarding controls selection, the authors wrote “The inclusion criteria were being born between 1989 and 2021 with no prior history of cancer diagnosis or recorded death before the age of 19, while allowing for cancer diagnosis at the age of 19 or older.” I guess that also controls did not emigrate before age 19, or otherwise you wouldn’t know whether they died or got a cancer diagnosis. However, since the controls cannot die before age 19 and some of the assessed perinatal characteristics are linked to childhood mortality (especially infant mortality) I wonder whether the decision to select controls only among children who didn’t die before age 19 could have led to a slight overestimation of the risk estimates, as these controls likely had more normal perinatal characteristics than the general population (that includes children who died during childhood). Moreover, I think that this sentence should be rephrased since it is not possible to check whether potential controls born in the more recent years are cancer-free at age 19 (or have not died by age 19) since they have not reached this age during the study period

According to the reviewer`s comment this part was rephrased as follows: “The inclusion criteria were being born between 1989 and 2021 with no known cancer diagnosis or recorded death before the age of 19, emigration or end of the study, while allowing for cancer diagnosis at the age of 19 or older. “

- Small for gestational age is defined as a birth weight below the 10th percentile for a specific gestational age, and these percentiles are usually estimated separately for boys and girls. This means that, in the general population, there are approximately 10% of children that are born SGA. However, when defining SGA as SD < -2, there will be a bit more than 2-3% of children defined as being born SGA (in Table 1 the authors reported that it was 4%, which is much lower than the definition of SGA). The same reasoning applies to LGA. Can the authors explain why they have defined SGA/LGA in a different way?

There are different definitions of SGA/LGA in the literature. We have used the definition that is mostly used in Sweden. This is based on ultrasonic estimates of foetal weight in normal pregnancies [Marsál K, Persson PH, Larsen T, Lilja H, Selbing A, Sultan B. Intrauterine growth curves based on ultrasonically estimated foetal weights. Acta Paediatr. 1996 Jul;85(7):843-8. doi: 10.1111/j.1651-2227.1996.tb14164.x. PMID: 8819552]. Using percentiles by gestational age and sex instead will systematically underestimate the magnitude of the growth restriction in infants born preterm. Another issue is which cut-off to use to define growth restriction. We could have used z-score <-1.28 (equivalent to the 10th percentile), but then a great proportion of the children below the cutoff would have been classified as SGA even if they are just normal but small. Using a more strict cut-off (-1.96) we are more likely to detect growth-restricted children.

- Information regarding child infections was obtained from the National Patient Register. Why did the authors use only the inpatient register and did not use the outpatient register data (that started in 2001)? Moreover, the authors should write in the text that they use inpatient care data, this is reported only in Table 1.

As stated within the methods under section Perinatal characteristics: “ … any child infection requiring specialist in-patient care since birth until the age of one year (yes, no)…”, we limited our analysis to infections recorded in in-patient care registries, as these reflect clinically significant cases requiring hospitalization and are more reliably diagnosed and documented. In contrast, outpatient data were not consistently available throughout the study period.

- The authors wrote that “Traditional case-control analyses using logistic regression do not allow estimation of time-varying associations” and therefore used Cox regression models to analyze their matched case-control data. However, I disagree with them because in a case-control study you can assess time varying associations, for example the effect of an exposure on childhood cancer risk during the first five years of life, by restricting the analyses to the cases diagnosed during the age period of interest and their matched controls. It is with time-varying covariates/exposures that one should use Cox regression when analyzing case-control data. I am not an expert on using Cox regression on case-control data but from the study below from Statistics in Medicine it seems that the naïve approach (analyzing the case-control data using the standard Cox regression model as if the data were obtained from a cohort study) could lead to an underestimation of the association. I therefore suggest to the authors to re-run the analyses using a conditional logistic regression model, as all the exposures that they have analyzed (perinatal factors) are not time varying.

Reference: Statistics in Medicine (2003) - Evaluation of Cox’s model and logistic regression for matched case-control data with time-dependent covariates: a simulation study

We thank the reviewer for this comment. We are aware that traditionally in case–control settings with matched controls and time-fixed exposures, conditional logistic regression is commonly applied. Cox regression falls under statistical techniques aimed at analysing “time-to-event” data and/or assessing the relationship between a given exposure and the occurrence of an outcome after a follow-up period among a cohort of individuals (ElHafeez et al., 2021; 10.1155/2021/1302811). In the present study, follow-up lengths, or time-to-event, vary considerably, with children born in the later part of the study period having significantly shorter follow-up compared to those born in the early years of the study. As such we wanted to address age at diagnosis (follow-up) as a potential factor influencing our tested association. While we agree that such patterns can also be explored through stratified logistic regression (restricting to cases diagnosed within specific age windows), such approaches require arbitrary partitioning of time and substantially reduce statistical power. In contrast, the Cox model allows us to use the full temporal information.

With the design of the present study, using loose matching (year of birth, county of birth and sex only), we don’t think that a matched analysis would be appropriate. In a simulation study, Kuo et al conclude: “Unmatched methods, e.g., unconditional logistic regression, are viable options for loose-matching data based on our findings. When the study design involves other complex features such as censoring and repeated measures, matching on a few demographic variables can be ignored if the confounding effect is not very large. Standard methods such as Cox regression and generalised estimating equation then can be readily applied.” [Kuo CL, Duan Y, Grady J. Unconditional or Conditional Logistic Regression Model for Age-Matched Case-Control Data? Front Public Health. 2018 Mar 2;6:57. doi: 10.3389/fpubh.2018.00057. PMID: 29552553; PMCID: PMC5840200].

Based on the comments from the other two reviewers, who supported our choice of statistical analyses—one specifically highlighting the use of Cox regression as a strength of the study—we have decided to retain Cox regression in the present study, unless the editor advises otherwise.

- The authors wrote “The models were not assessed for characteristics or categories with fewer than ten observations.” Please clarify whether with “ten observations” you mean ten cancer cases for a specific category or if with ten observations you meant a combination of cases and controls

We have rephrased the sentence as follows: “The models were not assessed for characteristics or categories when there were fewer than ten observations in a group of cases or controls.”

- Why did the authors add a quadratic term for BMI in the regression models? In the main analysis BMI was used as a categorical variable rather than a continuous one

Inclusion of the quadratic term of BMI was run as a sensitivity analysis; this was omitted from the manuscript.

- Did the authors have information regarding genetic syndromes? As these are linked to childhood cancer risk and could also be linked to some perinatal characteristics, is it possible to additionally adjust for genetic syndromes, at least for the most common ones such as Down syndrome, neurofibromatosis type 1, etc.?

We additionally obtained data on the diagnosis of some cancer predisposing syndromes, including Down syndrome, Neurofibromatosis type 1, Congenital malformation syndromes involving early overgrowth, and Von Hippel-Lindau syndrome. We conducted further analyses assessing the association between the diagnosis of these syndromes (yes/no) and the risk of childhood cancer. As suggested, we also performed a sensitivity analysis evaluating the association of all studied perinatal characteristics after excluding children diagnosed with any of these syndromes (n = 49). The exclusion of those children did not significantly affect our findings. These analyses are now added into the supplementary materials and described within the methods, results and discussion part of the manuscript.

- In the discussion the authors wrote “However, several studies, including a large cohort with over 11 thousand cases from Sweden, Denmark and Finland, reported null associations (65,66).”. However, references 65 and 66 refer to studies performed in the UK and the US. Please check that all references are cited correctly.

We have corrected the citation order of the references.

- I agree with the authors’ decision to not adjust for multiple comparisons. However, the authors should mention in the discussion section that because of the low number of cases and the several associations analyzed, it is possible that some of the reported statistically significant associations could be chance findings.

According to the comments and suggestions of 2 reviewers (reviewer 1 and 2), the following was added into the discussion of the manuscript within the limitations of the study part: “The statistical analysis of this study includes multiple comparisons, increasing the risk of false positives; however, given the limited statistical power, lowering the significance threshold could potentially overlook meaningful associations. Therefore, we chose to interpret the results cautiously in the context of existing literature rather than adjust for multiple comparisons, acknowle

---

## [Decision Letter · Decision Letter 1]

16 Mar 2026

Parental, pregnancy and neonatal characteristics during the perinatal period as potential risk factors for childhood cancer: FeToxCancer case-control study

PONE-D-25-24435R1

Dear Dr. Broberg,

We’re pleased to inform you that your manuscript has been judged scientifically suitable for publication and will be formally accepted for publication once it meets all outstanding technical requirements.

An invoice will be generated when your article is formally accepted. Please note, if your institution has a publishing partnership with PLOS and your article meets the relevant criteria, all or part of your publication costs will be covered. Please make sure your user information is up-to-date by logging into Editorial Manager at Editorial Manager® and clicking the ‘Update My Information’ link at the top of the page. For questions related to billing, please contact  and clicking the ‘Update My Information’ link at the top of the page. For questions related to billing, please contact  and clicking the ‘Update My Information’ link at the top of the page. For questions related to billing, please contact  and clicking the ‘Update My Information’ link at the top of the page. For questions related to billing, please contact billing support....

Kind regards,

Melvin Marzan, BSc, MSc TM, PhD

Academic Editor

PLOS One

Additional Editor Comments (optional):

Reviewers’ comments:

Reviewer’s Responses to Questions

**Comments to the Author**

Reviewer #2: All comments have been addressed

Reviewer #3: All comments have been addressed

2. Is the manuscript technically sound, and do the data support the conclusions?

Reviewer #2: Yes

Reviewer #3: (No Response)

3. Has the statistical analysis been performed appropriately and rigorously?

Reviewer #2: Yes

Reviewer #3: (No Response)

4. Have the authors made all data underlying the findings in their manuscript fully available?

Reviewer #2: No

Reviewer #3: (No Response)

5. Is the manuscript presented in an intelligible fashion and written in standard English?

Reviewer #2: Yes

Reviewer #3: (No Response)

Reviewer #2: The authors have sufficiently addressed all my previous comments.

Reviewer #3: (No Response)

.

Reviewer #2: No

Reviewer #3: No

---

## [Editor Report · Acceptance letter]

PONE-D-25-24435R1

PLOS One

Dear Dr. Broberg,

I’m pleased to inform you that your manuscript has been deemed suitable for publication in PLOS One. Congratulations! Your manuscript is now being handed over to our production team.

Lastly, if your institution or institutions have a press office, please let them know about your upcoming paper now to help maximize its impact. If they’ll be preparing press materials, please inform our press team within the next 48 hours. Your manuscript will remain under strict press embargo until 2 pm Eastern Time on the date of publication. For more information, please contact onepress@plos.org.

Kind regards,

on behalf of

Dr. Melvin Marzan

Academic Editor

PLOS One